# A Multilayer Functionalized Drug-Eluting Balloon for Treatment of Coronary Artery Disease

**DOI:** 10.3390/pharmaceutics13050614

**Published:** 2021-04-23

**Authors:** Hak-Il Lee, Won-Kyu Rhim, Eun-Young Kang, Bogyu Choi, Jun-Hyeok Kim, Dong-Keun Han

**Affiliations:** Department of Biomedical Science, CHA University, 335 Pangyo-ro, Bundang-gu, Seongnam 13488, Gyenggi, Korea; puma9307@naver.com (H.-I.L.); wkrhim@cha.ac.kr (W.-K.R.); subbi1009@naver.com (E.-Y.K.); bgchoi725@gmail.com (B.C.); gohom13@naver.com (J.-H.K.)

**Keywords:** cardiovascular disease, drug-eluting balloon, liposome, everolimus, restenosis

## Abstract

Drug-eluting balloons (DEBs) have been mostly exploited as an interventional remedy for treating atherosclerosis instead of cardiovascular stents. However, the therapeutic efficacy of DEB is limited due to their low drug delivery capability to the disease site. The aim of our study was to load drugs onto a balloon catheter with preventing drug loss during transition time and maximizing drug transfer from the surface of DEBs to the cardiovascular wall. For this, a multilayer-coated balloon catheter, composed of PVP/Drug-loaded liposome/PVP, was suggested. The hydrophilic property of 1st layer, PVP, helps to separate drug layer in hydrophilic blood vessel, and the 2nd layer with Everolimus (EVL)-loaded liposome facilitates drug encapsulation and sustained release to the targeted lesions during inflation time. Additionally, a 3rd layer with PVP can protect the inner layer during transition time for preventing drug loss. The deionized water containing 20% ethanol was utilized to hydrate EVL-loaded liposome for efficient coating processes. The coating materials showed negligible toxicity in the cells and did not induce pro-inflammatory cytokine in human coronary artery smooth muscle cells (HCASMCs), even in case of inflammation induction through LPS. The results of hemocompatibility for coating materials exhibited that protein adsorption and platelet adhesion somewhat decreased with multilayer-coated materials as compared to bare Nylon tubes. The ex vivo experiments to confirm the feasibility of further applications of multilayer-coated strategy as a DEB system demonstrated efficient drug transfer of approximately 65% in the presence of the 1st layer, to the tissue in 60 s after treatment. Taken together, a functional DEB platform with such a multilayer coating approach would be widely utilized for percutaneous coronary intervention (PCI).

## 1. Introduction

Coronary artery disease (CAD) is a serious disease that leads to death as the blood vessels are narrowed or almost closed by accumulation of fat or plaque, and the oxygen is not delivered to the heart muscles smoothly [1,2]. Percutaneous coronary intervention (PCI), a representative treatment for CAD, is a widely utilized method by inserting a guideline into the entrance to the coronary artery and expanding the constricted lesion with a stent, balloon, resection, or laser [3,4]. For the past fifty years, the medical devices for PCI, such as balloons, stents, drug-eluting stents (DESs) and bioresorbable vascular scaffolds (BVSs) have been developed [5,6,7,8,9,10]. Especially, local delivery method of proliferation inhibitor or immunosuppressant through DESs has shown excellence as an approach for successful CAD to reduce the rate of restenosis after PCI, one of the major drawbacks from bare metal stents [11,12]. However, there have been still some limitations, including late vascular reendothelialization, stent thrombosis, and restenosis due to the degradation of the permanent inserted foreign materials [13,14]. In addition, DESs were not ideal tools for CAD due to the potential risk of fracture and deformation with multidirectional mechanical forces [15]. To solve these problems, drug-eluting balloons (DEBs) have been developed and widely used in clinics. The cytostatic/cytotoxic drugs could be coated to DEBs to prevent the proliferation of smooth muscle cells (SMCs) with long-term patency without leaving an implant in the body [16,17,18]. Rapamycin is one of the most widely used drugs in interventional medical devices [19,20]. It is a macrolide derivative with immunosuppressive and anti-proliferative effects [21]. Two types of drugs, sirolimus (SRL) and everolimus (EVL), which inhibit the mechanistic target of rapamycin (mTOR) have been approved for immunosuppressant after transplantation in the United States and most European countries [22,23]. Among them, EVL has been applied as an immunosuppressive drug with the 40-O-(2-hydroxyethyl) derivate of SRL which enables sustained release in the blood vessel [24,25]. Major drawbacks of the balloon catheter have been drug loss during delivering to lesions due to highly viscous blood stream and low drug delivery rate to the tissues because of a short contact between drug-coated balloon and tissue of vascular lesions. To overcome these, herein, we suggested a multilayer-coated DEB system containing hydrophilic polymer and drug-loaded liposome (Figure 1). A biocompatible and biodegradable hydrophilic polymer, polyvinylpyrrolidone (PVP) was used to coat the 1st and 3rd layers of the balloon catheter to prevent drug loss during transition time and to facilitate separation of drug-loaded liposome (2nd layer) from the balloon catheter to the lesions in hydrophilic condition. Low molecular weight (Mw: 3.5 kDa) of PVP was utilized because PVP was approved as an excipient for human usages under 20 kDa under the Food and Drug Administration (FDA) [26]. Additionally, hydrophobic drug-loaded liposomes were introduced to deliver drug efficiently to the targeted region. Liposomes are artificial vesicles, mainly composed of amphiphilic phospholipids and formed in hydrophilic solvents [27,28,29]. The amphiphilic properties of liposome allow to carry both hydrophilic and hydrophobic materials to the targeted site. The hydrophilic drugs are entrapped within the aqueous inner core, and the hydrophobic drugs locate themselves between bilayers [30,31,32,33]. The EVL-loaded liposome (EVL + Lipo) structures were coated between the 1st and 3rd PVP layers to enable sustained release of drug in a targeted region after separation of PVP in the 3rd layer. The ratio of drug to lipid was optimized and the structures of hydrophobic drug (EVL)-loaded liposome was characterized with various tools. Additionally, multilayer coated surfaces of nylon tubes (a replacement of balloon catheter for efficient experiments in this study) were analyzed and interactions with human coronary artery smooth muscle cells (HCASMCs) were verified using various in vitro cell-based assays. The functionalities of each layer for efficient protection and transfer of drugs were demonstrated using fluorescence-based analysis at sink condition in vitro depending on time. The coating materials were not toxic and do not induce inflammatory cytokine in HCASMCs. Finally, an efficient drug transfer was simulated ex vivo using universal testing machine (UTM) in a condition similar to the pressure of the vascular tissue from the balloon catheter.

## 2. Materials and Methods

### 2.1. Materials

Nylon tube and polyvinylpyrrolidone (PVP, average Mw: 3.5 kDa) powder were purchased from Misumi (Tokyo, Japan) and Acros Organics (Geel, Belgium), respectively. Hydro soy phosphatidylcholine (HSPC) and 1,2-distearoyl-sn-glycero-3-phosphoethanolamine-N-[methoxy(polyethylene glycol)-2000] (mPEG2000-DSPE) were purchased from Avanti Polar Lipids (Alabaster, AL, USA). Cholesterol (Chol), 5(6)-Carboxyfluorescein (5(6)-FAM), and Nile red were purchased from Sigma (St. Louis, MO, USA). Everolimus (EVL) was provided by Osstem Cardiotech (Seoul, Korea). Phosphate-buffered saline (PBS) solution and fetal bovine serum (FBS) were purchased from Hyclone in GE life sciences (Marlborough, MA, USA). Trypsin EDTA, antibiotic-antimycotic, and Hoechst 33342 were obtained from Thermo Fisher Scientific (Waltham, MA, USA). A cell counting kit (CCK 8) was purchased from Dongin LS (Seoul, Korea). Human coronary artery smooth muscle cell (HCASMC) and smooth muscle cell growth medium-2 (SMGM-2) were purchased from Lonza (Basel, Switzerland). Lysotracker green DND-26 and 1,1’-dioctadecyl-3,3,3’,3’-tetramethylindotricarbocyanine Iodide (DiR’; DilC18(7)) were purchased from Invitrogen (Carlsbad, CA, USA). Fibrinogen from human plasma and albumin from human serum were purchased from Sigma-Aldrich (St. Louis, MO, USA), and porcine aorta vessel was provided from Daegu Gyeongbuk Medical Innovation Foundation (Daegu, Korea). An enzyme linked immunosorbent assay (ELISA) kit for interleukin-6 (IL-6) and interleukin-10 (IL-10) was purchased from R&D Systems Inc. (Minneapolis, MN, USA).

### 2.2. Preparation of Liposomes

Everolimus (EVL) was encapsulated in the liposome by thin-film hydration method without surfactants. EVL and liposome components (HSPC:Chol:mPEG2000-DSPE = 5.5:4:0.5) were dissolved in chloroform and then dried on a rotary evaporator under maintaining vacuum at 60 °C for 10 min. The evaporated thin film was resuspended in deionized water by rotating the flask at 120 rpm until the lipid membrane was completely hydrated. Then, the small unilamellar vesicles (SUVs) were formed using tip sonication for 10 min and bath sonication for 15 min, serially. Non-encapsulated drug was removed from the liposome dispersion by centrifugation at 3000 rpm for 15 min. The free drug could be aggregated and formed pellets due to their hydrophobicity [34]. The supernatant containing drug-loaded liposomes (Lipo + EVL) was collected and lyophilized (Laboratory floor model freeze-dryer, Ilshin FD8508, Korea) under vacuum condition at −80 °C for 24 h to concentrate for further experiments. The lyophilized Lipo + EVL powders were rehydrated in distilled water (DW) containing 20% ethanol for efficient coating processes to Nylon tubes.

### 2.3. Characterization of Drug-Loaded Liposomes

The size and surface charge of liposome formulations were measured using the dynamic light scattering (DLS, Zeta sizer, Malvern Panalytical, Malvern, UK) instrument after diluting liposomes in DW. Encapsulation efficiency (EE) is the ratio of the amount of drug encapsulated in liposomes to the amount of drug injected. After centrifugation at 3000 rpm for 15 min, the supernatant was collected to separate Lipo + EVL from free drug, and 0.9 mL of methanol was mixed with 0.1 mL of the supernatant and sonicated for 15 min [34]. The quantification of Lipo + EVL was determined using a high-performance liquid chromatography (HPLC, LC 1100, Agilent, Waldbronn, Germany) with Agilent Eclipse column (5 micron, 4.6 × 150 mm C −18) at 50 °C. The mobile phase of water, acetonitrile, and methanol (15:45:40) was flowed at a rate of 1.0 mL/min. The profile was monitored at 278 nm wavelength.

### 2.4. Cell Culture

Human coronary artery smooth muscle cells (HCASMCs) were cultured using smooth muscle cell growth medium (SmGM) containing 5% fetal bovine serum with growth factors and antibiotics in humidified incubator. All in vitro assays were performed with HCASMCs at passages 8.

### 2.5. Cell Viability and Intracellular Distribution of Liposome and EVL-Loaded Liposome

The cell viability incubated with various materials was evaluated using CCK 8 according to instructions received from the manufacturer, and the cell viability assay was performed in the media containing PVP to create the same conditions as the PVP of the 3rd layer was first released in the blood vessel. The concentration-dependent cytotoxicity of liposome and EVL-loaded liposome was measured after 24 h incubating with HCASMCs. To monitor intracellular distribution of liposomes, liposome was labeled with DiR through simple mixing for 1 h. Moreover, the lysosome and nucleus of HCASMCs were stained with Lysotracker and Hoechst, following the manufacturer’s instructions, respectively. The distribution of fluorescence-labeled liposomes was monitored using confocal microscopy (LSM 880, Zeiss, Oberkochen, Germany).

### 2.6. Ultrasonic Spray Coating

The ultrasonic spray coating instrument (SoniCoater, Noanix Co., Cheongju, Korea) was used to coat the balloon with coating materials. Herein, a Nylon tube with same composition as the balloon was utilized for efficient experiments. Before coating processes, the Nylon tube was washed with methanol and acetone, and placed on a mandrel. PVP dissolved in tetrahydrofuran (THF, 1 mg/mL) and EVL-loaded liposome dissolved in 20% ethanol (2 mg/mL) were sequentially sprayed. All coating processes run at a flow rate of 0.05 mL/min. For coating 750 ug EVL-loaded liposomes, nine-repeated coating was performed with 150 μL once. Finally, PVP solution in acetone (1 mg/mL) was stacked on the EVL-loaded liposome-coated 2nd layer.

### 2.7. Analysis of Surface Properties

The morphology of the cross-section of the composites was observed by field emission-scanning electron microscopy (FE-SEM, JEOL, Tokyo, Japan). The specimen was fractured after immersing in liquid nitrogen. To visualize the surface of Nylon tubes coated with PVP and PVP/Lipo + EVL, 5(6)-Carboxyfluorescein (5(6)-FAM)-stained PVP and Nile red-labeled liposome were utilized. The surface topography and roughness value were measured using an atomic force microscopy (AFM, Multimod IVa, Bruker, Billerica, MA, USA) under a tapping mode at room temperature. The root mean square (RMS) values were calculated for the average of the three least-distinct spots (40 × 40 μm) on different substrates. Attenuated total reflectance-Fourier transform infrared spectroscopy (ATR-FTIR, Spectrum two, PerkinElmer, Waltham, MA, USA) was used to detect IR transmittance of surface at a range of 4000 to 650 cm^−1^. The differences in surface atomic concentration were compared with X-ray photoelectron spectroscopy (XPS, PHI5000 VersaProbe, Ulvac-PHI, Chigasaki, Japan). The hydrophilicity of the surface was evaluated based on the contact angle (θ) of a water droplet with the multilayer coated films using optical bench-type contact angle goniometry (Crest Technology Co., Sterling, MA, USA). A drop of 20 μL of deionized water was placed on coated surface, and then the water contact angle was evaluated within 10 s at different regions of the specimens.

### 2.8. Drug Release and Separation of Coated Layers

For in vivo mimicking environments, the condition for sink solution (0.05% Tween 20 in PBS solution) was created at 37 °C with 300 rpm, which could monitor drug release from the multilayer-coated Nylon tubes and the separation of each layer under the similar condition with blood vessels. The loading and total amounts of releasing EVL were quantified using HPLC. To determine the amount of released EVL from multilayer-coated Nylon tube, the standard curve was prepared according to drug concentration in the range of 0.5~1000 μg/mL. The cumulative amount of drug release was checked for 28 days. To optimize coating condition of PVP onto Nylon tubes, Nylon tubes were coated 75, 100, 150, and 200 times with 5(6)-FAM labeled PVP. Each sample was exposed with stirred sink condition at a rate of 300 rpm for 10 s, and the amount of remaining fluorescence labeled-PVP onto Nylon tube was measured using a fluorescence in vivo imaging system (FOBI; CELLGENTEK, Cheongju, Korea). In addition, to verify time-dependent separation of the 2nd and 3rd layers, PVP (1st layer)-doped Nylon tubes were coated with Nile red labeled liposomes and 5(6)-FAM labeled PVP, sequentially. After soaking in a solution of sink condition with stirring at 300 rpm for 10, 30, and 60 s, the fluorescence intensity was measured using FOBI.

### 2.9. Enzyme-Linked Immunosorbent Assay (ELISA)

The expression level of inflammation-related factors (pro-inflammatory cytokine: IL6 and anti-inflammatory cytokine: IL-10) in HCASMC incubated with coating materials were analyzed using the Quantikine™ ELISA kit (R&D Systems, Minneapolis, MN, USA). To determine the appropriate cellular environment for performing inflammation assays, the intracellular toxicity of coating materials over time was evaluated, and a CCK 8 assay kit was utilized to measure cell viability and a live/dead assay was further conducted to visualize the healthy cells as followed by the manufacturer’s instructions. Briefly, after incubating with coating materials for 6 and 24 h, the cells were rinsed three times with 1× PBS solution and incubated with Calcein-AM and EthD-1 at room temperature in the dark. After 30 min, the cells were rinsed with PBS solution and observed using a fluorescence inversion microscopy (CKX53, OLYMPUS, Tokyo, Japan). The level of IL-6 and IL-10 was evaluated using ELISA at 6 h after incubation. To evaluate the effects of coating materials to reduce release of pro-inflammatory cytokine, the 100 ng/mL of lipopolysaccharide (LPS) were pretreated to the cells. The process of ELISA was performed according to instructions received from the manufacturer.

### 2.10. Hemocompatibility Test

Human plasma fibrinogen and human serum albumin were used as standards to evaluate the protein adsorption on the composite films. The multilayer coated film was pre-wetted with PBS solution for 1 h at 37 °C, and incubated with fibrinogen solution (0.2 mg/mL) and albumin solution (3.0 mg/mL) for 2 h at 37 °C, respectively. The multilayer-coated film was washed with distilled water, and the amount of protein was quantified using a micro-BCA assay kit as followed by the manufacturer’s instructions. To evaluate platelet attachment onto the coating materials, all coating materials were hydrated in 1 mL PBS solution at 37 °C for 1 h, and incubated with the concentrated platelet (3.5 × 10^4^ platelets/mL, Nambu Blood Institute, Seoul, Korea). After serial dehydration with 50, 60, 70, 80, 90, and 100% ethanol, platelet adhesion was checked with SEM. The lactate dehydrogenase (LDH) assay was also performed to quantify platelet adhesion onto the coating materials as described in the manufacturer’s instructions (LDH cytotoxicity detection kit, TAKARA, Tokyo, Japan).

### 2.11. Ex Vivo Drug Transfer Study

The ex vivo drug transfer rate was evaluated with porcine aortic vessels using universal testing machine (UTM, TO-101, Test one, Busan, Korea). Briefly, the porcine aortic vessel was placed on the test mold and a multilayer-coated film was placed on the surface of the opposite site. Then, two molds were compressed for 10 and 60 s with 0.9 bar, equal to the pressure applied to the vascular tissue through the balloon during the treatment, to attach each other. After separating from the mold, each sample was immersed in chloroform to quantify the amount of drug retention. The Nile red-labeled liposome was utilized to monitor drug transfer onto tissues using FOBI.

### 2.12. Statistical Analysis

Values are presented as mean ± SD (*n* = 3). All statistical analyses were performed using GraphPad Prism 7 (GraphPad Software, San Diego, CA, USA). Differences between groups were assessed using one-way analysis of variance (ANOVA) with Tukey’s multiple comparison post-test and *p* values below 0.05 were considered as statistically significant (* *p* < 0.05; ** *p* < 0.01; *** *p* < 0.001; **** *p* < 0.0001).

## 3. Results

### 3.1. Liposome Characterization

To optimize the ratio of lipid and drug to maximize drug loading, the various ratios of lipids and drugs were adjusted (Figure 2A). As the ratio of lipid increased to 3000:100, the amount of loaded drug increased, whereas the yield of drug encapsulation decreased according to increasing the ratio of lipid over 1000:100. The drug-to-lipid ratio of 2000:100 was chosen for further experiments. The stability of drug-loaded liposomes was verified with changing of size and zeta potential in various conditions (Figure 2B,C). As a high concentration of lipid-drug complex is required to coat drug to the surface of a balloon, rehydration after the lyophilization process is necessary, and 20% ethanol was added to the liposome solution for efficient coating to the Nylon tube. To confirm stability of drug-loaded liposome during these processes, the size and zeta potential were compared. The results showed no significant difference in size and zeta potential (Appendix A) compared to optimized drug-loaded liposome structures. The TEM images displayed liposome structures with round shapes at a size range of 70 to 100 nm (Figure 2D). Additionally, encapsulated drugs were stable during 28 days without significant degradation (Figure 2E).

### 3.2. Cell Viability and Intracellular Distribution of Liposome and EVL-Loaded Liposome

The cytotoxicity of liposomes and EVL-loaded liposomes was analyzed in HCASMCs. The high concentration of liposome (~1000 μg/mL) exhibited no cytotoxicity without EVL, and EVL-loaded liposomes showed significant cytotoxicity over 1000 μg/mL due to cytotoxic properties of drug (Figure 3A,B). The endocytic pathway of EVL-loaded liposome internalization into cells was monitored using colocalization of DiR-labeled liposome with fluorescent-stained lysosome (Figure 3C). Liposome could facilitate internalization and activation of drugs in the cell.

### 3.3. Analysis of Multilayer-Coated Surface

To simulate functionalities of multilayer coating for drug-eluting balloon system, the Nylon tube was utilized as a replacement for balloon catheters. The property and morphology of each layer were analyzed using various methods, including SEM, fluorescence microscopy, AFM, ATR-FTIR, and XPS. The surface was smooth after coating of the 1st layer with PVP and some roughness appeared after the EVL-loaded liposome was coated. After coating with PVP as a 3rd layer, the roughness was shown to be disappeared (Figure 4A). The coating of PVP and EVL-loaded liposome onto the Nylon tube was visualized using 5(6)-FAM-labeled PVP and Nile red-stained liposome, respectively (Figure 4B). The roughness of the coated surface is an important factor to inhibit protein absorption in blood vessel environment. Through AFM analysis, it could be proved that surface morphology of the coated surface was changed as the coating layers were stacked (Figure 4C). The value of root mean square (RMS) increased with the same tendency (Table 1). Additionally, the changes of molecular interaction for coated surfaces were characterized using ATR-FTIR with the shift of broadening of the peak in the transmission spectra. The spectra of OH bond (at 3352~3404 cm^−1^), C-H asymmetric stretching vibration (at 2955 and 1661 cm^−1^), and C = O amide stretching vibration and CH_2_ bending vibration (at 1424 and 1291 cm^−1^, respectively) were measured after coating the 1st PVP layer onto Nylon tubes. With coating the 2nd layer using EVL-loaded liposome, the C = O ester binding (at 1736 cm^−1^) and the PO antisymmetric stretch (at 1232 cm^−1^) were detected, which diminished as the 3rd layer, PVP, was coated with appearance of C = O amide stretch (at 1657 cm^−1^) clearly (Figure 4D). The clear coating of each layer was also verified using XPS which allows the determination of elements and chemical compositions of the coated layer. As shown in Table 1, the atomic percentages of carbon (C), oxygen (O), nitrogen (N), and phosphorous (P) contents for bare Nylon tube, PVP, PVP/Lipo + EVL, and PVP/Lipo + EVL/PVP were measured. For PVP coating, the percentage of only N1s showed a slight increase compared to bare Nylon tubes. Once the EVL-loaded liposome (Lipo + EVL) was coated in a second layer, a P2*p* value derived from liposome was newly detected with N1s diminishing due to being overall coated with Lipo + EVL. Finally, the results of the water contact angle proved that each layer was well coated with hydrophilic materials (Table 1).

### 3.4. Functionalities of PVP Layers

To demonstrate the role of the PVP coated 1st layer, the drug release profile was monitored for 28 days. The loading amount of EVL, 750 μg (3 μg/mm^2^) for clinical treatment, was similar depending on the presence or absence of the PVP layer (Figure 5A), and the release behavior of EVL contents exhibited a two-phase release pattern including an initial burst release for 7 days followed by sustained release for up to 28 days. Interestingly, the large amount of EVL was initially released and the cumulative amount of EVL release increased as the PVP-coated 1st layer was present. The 78.2% of drug was released by coating PVP on the first layer (Figure 5B), which indicated the role of the hydrophilic PVP layer for an efficient separation of coated layer in the hydrophilic condition of blood vessel. The time-dependent separations of both the 2nd Lipo + EVL and 3rd PVP layers were monitored via 5(6)-FAM-labeled PVP and Nile red-stained liposomes. In order to provide a similar environment for blood flow in blood vessel in vitro, the set up for sink condition with stirring was established as shown in Figure 5C. It needed 100 repetitive coatings to determine the optimal number of PVP coating processes separated from the Nylon tube within 10 s. The coated PVP layers were almost separated after 10 s in the solution of the sink condition with stirring, as expected (Figure 5D). Moreover, the layer of Lipo + EVL was well coated in the initial 10 s and started to be released after separation of the 3rd layer for the next 50 s (Figure 5E).

### 3.5. Anti-Inflammatory Property of the Coating Materials

CCK-8 and live and dead assays were performed to evaluate time-dependent cytotoxicity of coating materials, PVP, liposome, and EVL-loaded liposome. The results demonstrated no significant cytotoxicity of all coating materials during 6 h incubation, but only 40% cells were alive at 24 h in the group of EVL-containing material due to the cytotoxic property of drug (Figure 6A). The similar tendency was monitored in live and dead assay using fluorescence (Figure 6B). To evaluate the effect of coating materials on the production of inflammatory cytokines, the protein expression levels of representative pro-inflammatory and anti-inflammatory cytokines were measured using ELISA. As shown in Figure 6C, the production of pro-inflammatory cytokines, IL6, in HCASMC was effectively reduced after serial coating with PVP and liposome, and dramatically inactivated in the Lipo + EVL-treated group especially, compared to the group for bare Nylon tube. Interestingly, all coating materials effectively decreased the level of IL-6 after induction of LPS, and the level of anti-inflammatory cytokine, IL10, displayed similarly in all groups (Figure 6C).

### 3.6. Hemocompatibility

The albumin is a passivating protein, whereas fibrinogen supports platelet adhesion. The BCA results showed that protein adsorption did not increase with multilayer-coated materials (Figure 7A,B). Depending on the coating materials, it was difficult to find the differences on the platelet adhesion tendency in SEM images (Figure 7C). From the results of LDH assay, the platelet adhesion was slightly reduced after coating with PVP, PVP/Lipo and PVP/Lipo + EVL serially, compared to bare Nylon tubes (Figure 7D).

### 3.7. Ex Vivo Study

The ex vivo drug delivery on the tissue from the coated materials was simulated using the customized UTM system (Figure 8A). To test drug transfer from coating surface to the tissues, the tissue of porcine aortic vessel and Nile red-loaded liposome was utilized, and the differences of fluorescent intensity were monitored with FOBI instruments. For the initial 10 s, the liposome layer was not separated due to the presence of the 3rd layer, and the liposome layer was separated more efficiently in the group with the 1st layer until 60 s (Figure 8B). The drug transfer rate was calculated using fluorescence intensity of transferred Nile red-loaded liposomes. More liposome layer was transferred to the tissues (~65%) in the presence of the 1st layer, which means that the 1st layer, PVP, helped to be separated efficiently in the hydrophilic conditions compared to the condition for the absence of the 1st layer (Table 2).

## 4. Discussion

Despite consistent developments of various technologies for PCI treatments, the stent types with permanent implantation have been limited for their late vascular reendothelialization, stent thrombosis, and restenosis due to the long-lasting foreign materials. Especially, the DESs have limitations in drug delivery efficiency because they contact only 15% of the vessel wall in the lesions, where drug elution occurs [35]. Also, the DESs induce excessive intimal hyperplasia and do not enable adaptive remodeling [36]. The implanted stents with restenosis and thrombosis are difficult to reopen. To overcome these drawbacks, DEBs have been recently developed for coronary angioplasty due to efficient drug delivery with reducing in-stent restenosis and removing a catheter from the body [37,38]. Using a DEB, it has become possible to coat the surface of the artery wall homogeneously with drug in the lesions. The ideal DEBs have been defined for achieving a uniform coating of drug that maximizes drug retention with inhibiting proliferation of SMCs. Due to the short drug treatment time, it is required for fast detachment from the balloon and higher efficient drug delivery to the tissue. Paclitaxel, the 1st-generation drug to treat vascular diseases, can inhibit the proliferation of SMCs with short time treatment [39,40]. However, treatment with alternative drugs has been introduced due to various problems, including the delayed reendothelialization and vascular healing after paclitaxel administrations with randomized inhibition of both endothelial cells (ECs) [41,42]. As a second generation of drug for DEBs, a type of rapamycin, sirolimus (SRL) and everolimus (EVL), has been investigated for use recently. The migration of endothelial cells is not as thoroughly inhibited by sirolimus. Besides, rapamycin shows anti-inflammatory effects to the tissue after its antiproliferative property. Therefore, a result for the use of rapamycin should be more reasonable to the vessel wall in case of overcoming the limitations of the poorer transfer rate compared to paclitaxel. Moreover, long persistence is necessary in tissue due to the reversible binding to the mammalian target of rapamycin receptors [43]. The aim of our study was to load drug onto a balloon catheter without drug loss during transition time and with maximizing drug transfer from the surface of DEBs to the cardiovascular wall. For this, it was suggested to use the multilayer-coated balloon, composed of three different layers, PVP/Drug-loaded liposome/PVP. The hydrophilic property of the 1st layer, PVP, helps to separate the drug layer in blood vessel, and the 2nd layer with drug-loaded liposome facilitates drug encapsulation and sustained release to the targeted lesions during inflation time. Additionally, the 3rd layer with PVP can protect the inner layer during transition time for preventing drug loss. To encapsulate hydrophobic drug, EVL, biocompatible liposome was used. Liposome can encapsulate the hydrophobic drug between bilayers without surfactants via simple mixing during thin-film hydration process for liposome formation [27,44]. The free drugs that cannot enter the liposomes are aggregated in hydrophilic condition and separated through centrifugation [45]. By controlling the ratio of liposome to drug, the amount of loaded drug with good encapsulation yield can be optimized. In order to coat large amounts of EVL-loaded liposome (Lipo + EVL) onto the surface of PVP-coated Nylon tubes more efficiently with ultrasonic spray, the Lipo + EVL formulations were concentrated with rehydration in 20% ethanol solution after lyophilization [46]. The structure of liposomes hydrated with deionized water containing 20% ethanol was stable with similar size and zeta potential as optimized Lipo+EVL. The endocytic pathway of drug-loaded liposomes internalization into the cell was demonstrated using fluorescence-based assay with fluorophore labeled liposome and cell organelles.

In this study, the Nylon tube (same components as balloon catheter) was utilized for efficient experiments. It is necessary to coat the surface of a balloon catheter smoothly to prevent protein adsorption in vivo system. Each coated surface of Nylon tube was verified using SEM, fluorescence microscopy, AFM, ATR-FTIR, and XPS. To inhibit platelet adhesions during treatments, a low level of surface roughness and hydrophilicity are important parameters [47,48,49]. The surface roughness of coated surfaces, RMS value, was obtained as 28.483 nm in the surface of bare Nylon tube, 79.304 nm in the 1st PVP coated layer, 135.18 nm in the 2nd Lipo + EVL-coated layer onto PVP coated surface, and 162.80 nm in the additional 3rd PVP coated layer using AFM analysis. As the coating layer was accumulated, the roughness tended to increase, but it was less than 2 μm, which is known to be attached easily by platelet. It can be seen by measuring the water contact angle that PVP and liposome increase the hydrophilicity of the surfaces which is known to interact with water molecules, making the balloon catheter biocompatible by inhibiting nonspecific protein adsorption. With a different value in N1s and P2p, it can prove that PVP and liposome are well coated. Especially, P2p value was detected only in the liposome-coated tube. Time-dependent separation of each layer is considered important for efficient treatment using a DEB system in clinics. The short treatment time and continuous effects are great advantages that can be expected from DEBs [50]. The cumulative release of the entire drug increased by the 1st PVP layer was introduced to improve the delivery ability of the drug by facilitating separation from the balloon catheter in the hydrophilic lesion site. Besides, the 3rd PVP layer protects the Lipo + EVL for the initial 10 s when the catheter reaches the lesions, and helps the drug to be released in the targeted site. The secretion of pro-inflammatory cytokines was reduced by all coating materials, and especially in the presence of drug, it indicated significant decreases even in the condition of inflammation induction by LPS [51]. IL-6 causes acute phase response with recruiting neutrophil and monocytes at the site of inflammation [52,53]. The expression level of IL-6 involved in the process of restenosis after vascular stent implantation has been reported [54]. Our results demonstrated the suppression of inflammatory responses by coating materials and EVL as described above. Protein adsorption is known as the first event that is caused by the complex interactions between the blood and the surfaces of catheters due to hydrophobic interaction and surface roughness [55]. Especially, the level of fibrinogen adsorption onto the coated materials was investigated to confirm the compatibility for further applications of coating materials because fibrinogen induces platelet adhesion and clotting with converting to fibrin by thrombin [56]. The BCA result showed that both albumin and fibrinogen did not significantly increase depending on the coating materials, which were consistent with platelet adhesion images by SEM. The changes in platelet adhesion according to coating materials were measured using LDH assay. Since the LDH level increases with the platelet activation, platelet adhesion onto the coating materials can be indirectly quantified by the level of LDH [57,58]. As expected, all hydrophilic coating materials tend to reduce platelet adhesion compared to hydrophobic bare Nylon tubes. Finally, to confirm the feasibility of further applications of multilayer-coated strategy as a DEB system, ex vivo drug transfer study was assessed using UTM. Time-dependent separation of layers was monitored in the pressure of 0.9 bar, similar as the pressure applied to the vascular tissue through the balloon during the treatment. The results display that the 3rd layer was separated after 10 s of contact, and the 2nd layer of drug-loaded liposome was released to the tissue in opposite until after 60 s. In addition, drug transfer rate increased the group with a 1st PVP layer.

## 5. Conclusions

A multilayer-coated functional DEB system was suggested in this study. The biocompatible and hydrophilic PVP was used to minimize drug loss on the surface of the balloon during transition time, and quickly be separated in a hydrophilic environment for efficient drug delivery to lesions. Moreover, liposome was utilized as a carrier for loading hydrophobic drug without crystallization in a hydrophilic environment, and for efficient drug delivery. Using a smooth coating technology, functional three different layers were coated efficiently. The coating materials did not induce an inflammatory response, and inhibited the release of pro-inflammatory cytokine after LPS induction in the presence of drug, especially. In addition, hydrophilic properties of multilayer-coated surface inhibited platelet adhesion, which prevent clot formation after treatment. The result of ex vivo experiments with our customized set-up demonstrated the role of PVP for the protection of Lipo + EVL in the initial 10 s and the property of liposome for efficient drug transfer to the tissue in 60 s after treatment, respectively. Based on this study, this multilayer coating approach can be extensively applied as an advanced DEB platform for the treatment of coronary artery diseases.

## Figures and Tables

**Figure 1 pharmaceutics-13-00614-f001:**
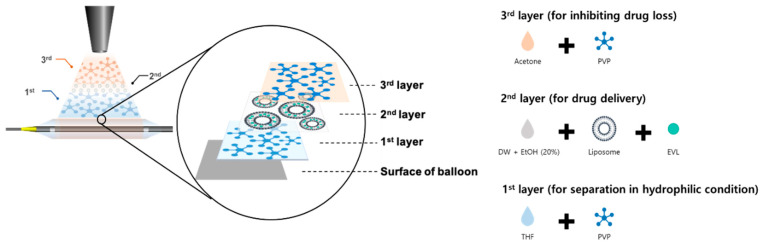
Schematic illustrations for a multilayer functionalized drug-eluting balloon (DEB) system.

**Figure 2 pharmaceutics-13-00614-f002:**
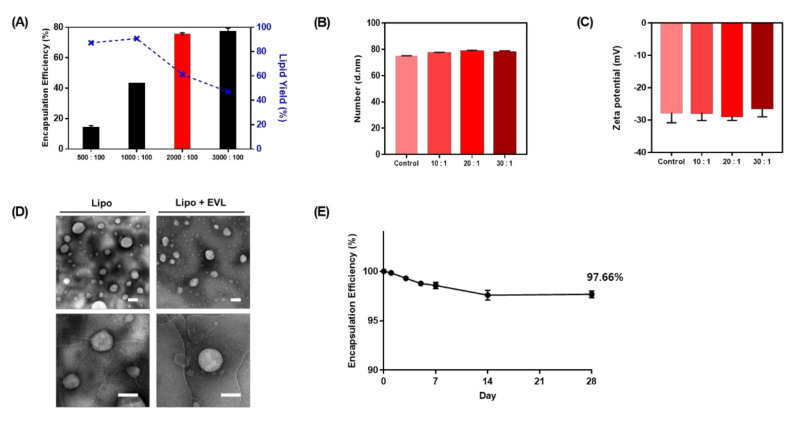
Characterization of liposome. (**A**) Analysis of encapsulation efficiency and drug encapsulation yield according to the ratio of lipid to drug. The ratio indicates Lipid:Drug. (**B**) Size and (**C**) Zeta potential of drug-loaded liposome depending on the ratio of lipid to drug. (**D**) TEM images of Lipo and Lipo + EVL (Scale bars equal to 100 nm). (**E**) The stability of Lipo + EVL for 28 days in PBS solution. Values are presented as mean ± SD (*n* = 3). Lipo: Liposome; Lipo + EVL: Everolimus (EVL)-loaded liposome.

**Figure 3 pharmaceutics-13-00614-f003:**
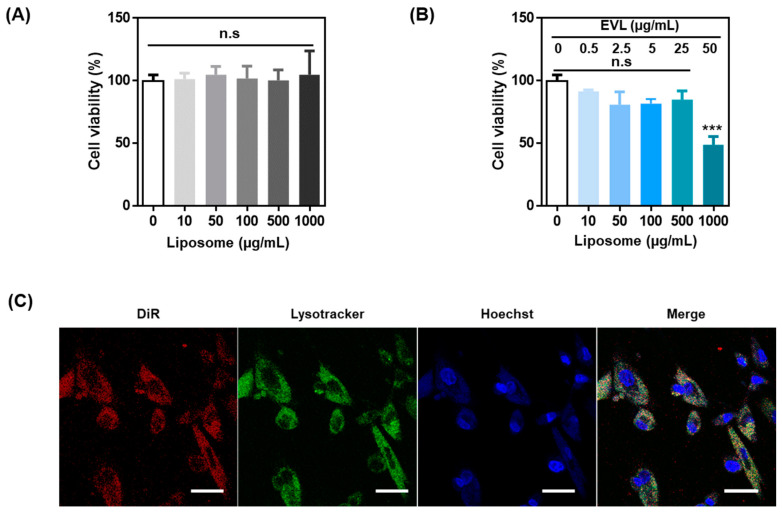
In vitro study using liposome and EVL-loaded liposome. Cell viability of (**A**) liposome and (**B**) EVL-loaded liposome. (**C**) Confocal images for interaction between DiR-labeled liposome and HCASMCs. The lysosome and nucleus of HCASMCs were stained by lysotracker (green) and Hoechst (blue), respectively (Scale bars equal to 40 μm). Values are presented as mean ± SD (*n* = 3) and statistical significance was obtained with one-way analysis of ANOVA with Tukey’s multiple comparison post-test (* *p* < 0.05; ** *p* < 0.01; *** *p* < 0.001). HCASMCs: Human Coronary Artery Smooth Muscle Cells.

**Figure 4 pharmaceutics-13-00614-f004:**
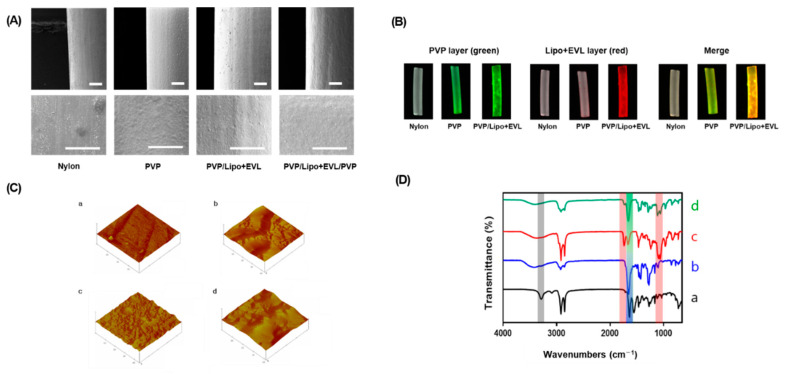
Characterization of PVP, PVP/Lipo + EVL, and PVP/Lipo + EVL/PVP-coated surface of Nylon tubes. (**A**) SEM images of the coated surface (Scale bars equal to 400 μm). (**B**) Fluorescence images of the coated surface using FOBI. GreenL 5(6)-Carboxyfluorescein-stained PVP, Red: Nile red-labeled liposome. (**C**) AFM images for surface roughness evaluation. (**D**) ATR-FTIR analysis for Nylon tube surface coated with PVP, PVP/Lipo + EVL, and PVP/Lipo + PVP/PVP. (**a**) Bare Nylon tube, (**b**) PVP-coated Nylon tube, (**c**) PVP/Lipo + EVL-coated Nylon tube, and (**d**) PVP/Lipo + EVL/PVP-coated Nylon tube.

**Figure 5 pharmaceutics-13-00614-f005:**
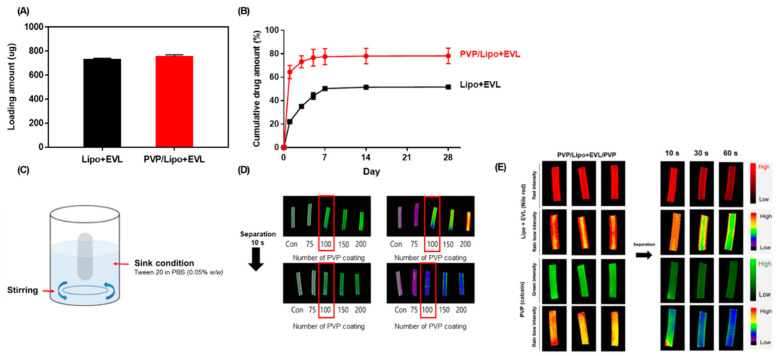
Functionalities of PVP and Lipo+EVL-coated layers. (**A**) Loading amount of drug onto Nylon tube depending on 1st layer with PVP. (**B**) Comparative study on drug release of Lipo + EVL-coated Nylon tube (black) and PVP/Lipo + EVL-coated Nylon tube (red). (**C**) The set-up for in vitro experiments to mimic blood flow in the blood vessel. (**D**) Optimization of coating condition for PVP and separation of the 1st PVP layer for 10 s. (**E**) The fluorescence images for time-dependent separation of coated layer for 10, 30, and 60 s in sink condition. Red: Nile red-labeled liposome, Green:5(6)-carboxyfluorescein-labeled PVP. Values are presented as mean ± SD (*n* = 3).

**Figure 6 pharmaceutics-13-00614-f006:**
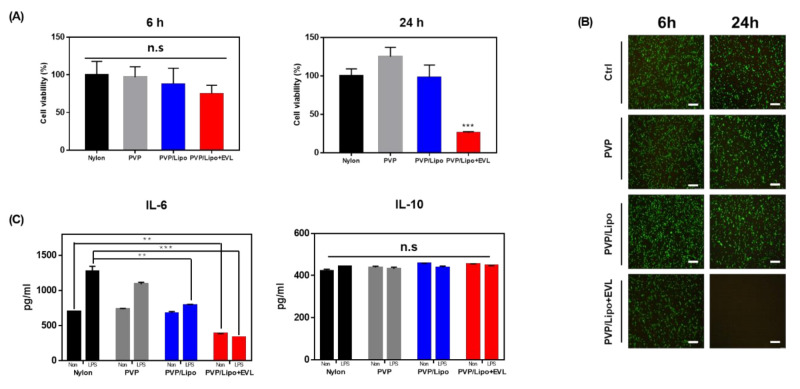
Cell viability and inflammation study. (**A**) Time-dependent cytotoxicity of PVP, PVP/Lipo, and PVP/Lipo + EVL. (**B**) Live and dead assay to visualize cytotoxicity of coating materials (Scale bars equal to 400 μm). (**C**) ELISA assay to confirm level of pro-inflammatory (IL-6) and anti-inflammatory (IL-10) factors in HCASMCs incubating with bare Nylon tube, PVP, PVP/Lipo, and PVP/Lipo + EVL. Values are presented as mean ± SD (*n* = 3) and statistical significance was obtained with one-way analysis of ANOVA with Tukey’s multiple comparison post-test (n.s: non significancy; ** *p* < 0.01; *** *p <* 0.001).

**Figure 7 pharmaceutics-13-00614-f007:**
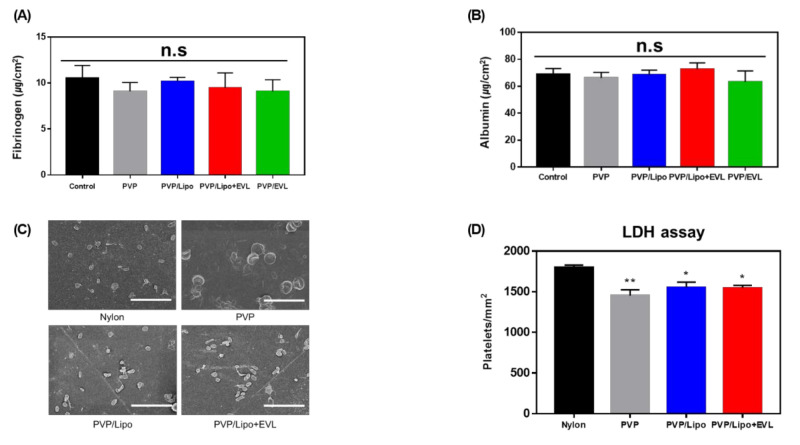
Hemocompatibility test. The degree of protein adsorption of (**A**) Fibrinogen and (**B**) Albumin depending on various coated layers (Control means bare Nylon tubes). (**C**) SEM images for platelet adhesion incubating with coated materials (Scale bars equal to 20 μm). (**D**) LDH assay to evaluate platelet adhesion onto coated materials. Values are presented as mean ± SD (*n* = 3) and statistical significance was obtained with one-way analysis of ANOVA with Tukey’s multiple comparison post-test (* *p* < 0.05; ** *p* < 0.01).

**Figure 8 pharmaceutics-13-00614-f008:**
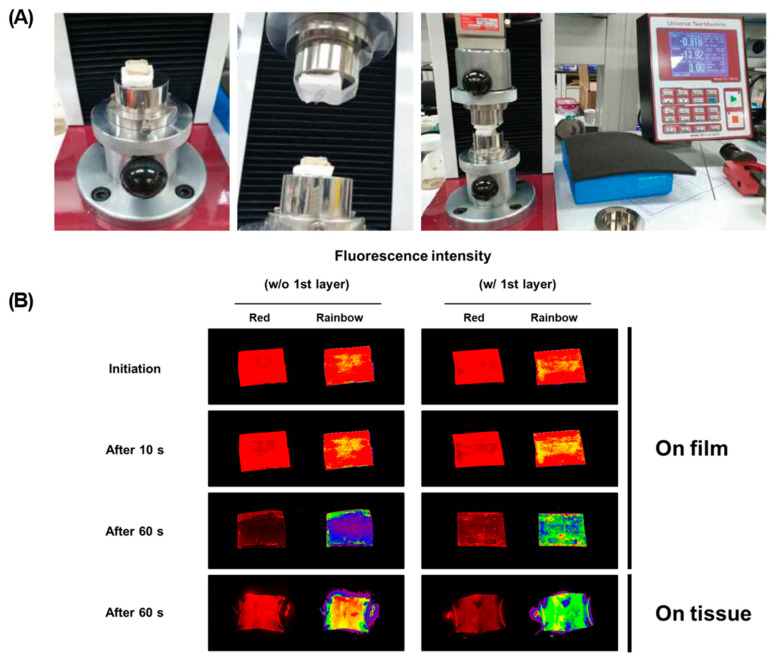
Ex vivo study for drug transfer on the tissue from multilayer-coated film. (**A**) Customized set-up to simulate drug transfer on the tissue from multilayer-coated film. (**B**) The fluorescence images for drug transfer to the tissues according to the presence of the 1st layer PVP (Red: Nile red-labeled liposome; The size of film is 20 mm × 12.48 mm).

**Table 1 pharmaceutics-13-00614-t001:** Summary of XPS atomic concentration, contact angle, and surface roughness from AFM for various treated Nylon tubes.

Sample	XPS Atomic Concentration (%)				Contact Angle (°)	RMS Roughness (nm)
	C1s	O1s	N1s	P2p		
Nylon	85.09	10.47	4.43	-	92.46	28.483
PVP	81.95	10.71	7.35	-	60.2	79.304
PVP/Lipo+EVL	84.7	12.09	1.71	1.5	19.26	135.18
PVP/Lipo+EVL/PVP	81.95	10.51	7.54	-	33.83	162.80

PVP: polyvinylpyrrolidone; Lipo + EVL: Everolimus-loaded liposome; XPS: X-ray photoelectron spectroscopy; RMS: Root mean square.

**Table 2 pharmaceutics-13-00614-t002:** Comparisons of drug transfer rate from ex vivo experiments.

Condition	Drug Transfer Rate (%)	Drug Remaining (%)	Drug Loss (%)
Lipo+EVL/PVP (w/o 1st layer)	50.19	33.78	14.39
PVP/Lipo+EVL/PVP (w/1st layer)	65.43	16.55	17.43

## Data Availability

Data are contained within the article and Appendix A.

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
