# Peer review of "A Multilayer Functionalized Drug-Eluting Balloon for Treatment of Coronary Artery Disease"

_pharmaceutics, 2021, doi:10.3390/pharmaceutics13050614_

Round 1

Reviewer 1 Report

The manuscript entitled “A Multilayer Functionalized Drug-Eluting Balloon for Treatment of Coronary Artery Disease” is original and has a significance for the scientific community. Experimental and theoretical approach to the discussed problem is good presented in the manuscript. Obtained results are reliable and supported by the data collected. The manuscript is easy to read and the arguments are described in a logical and understandable way. 

In order to improve the manuscript, the following suggestion can be taken into account by the authors:

  1. How was determined the absence of free Everolimus (EVL) in the supernatant (lines 115, 116)? Please, add this information to the Manuscript.

After these minor corrections, I highly recommend to accept this manuscript for publication.

Author Response

Response to the Referees’ comments

First of all, we would like to thank the reviewers for a thorough reading of the manuscript and helpful comments and suggestions in details. We have revised our manuscript based on the comments by reviewers, and we believe the manuscript is now significantly improved. The followings are the point-by-point responses to the reviewers’ comments.

Comments from the editors and reviewers:

Reviewer 1

Overall Summary: The manuscript entitled “A Multilayer Functionalized Drug-Eluting Balloon for Treatment of Coronary Artery Disease” is original and has a significance for the scientific community. Experimental and theoretical approach to the discussed problem is good presented in the manuscript. Obtained results are reliable and supported by the data collected. The manuscript is easy to read and the arguments are described in a logical and understandable way. 

In order to improve the manuscript, the following suggestion can be taken into account by the authors:

After these minor corrections, I highly recommend to accept this manuscript for publication.

Comments 1: How was determined the absence of free Everolimus (EVL) in the supernatant (lines 115, 116)? Please, add this information to the Manuscript.

Response 1: We appreciate for the reasonable comment. We isolated free drug using centrifugation as a followed reference (Abud, M.B.; Louzada, R.N.; Isaac, D.L.C.; Souza, L.G.; Dos Reis, R.G.; Lima, E.M.; de Ávila, M.P. In vivo and in vitro toxicity evaluation of liposome-encapsulated sirolimus. Int. J. Retin. Vitr. 2019, 5, 1-10). The free drug could be aggregated and formed pellet due to their         hydrophobicity. And drug-encapsulated liposome was well dispersed in supernatant. The sentence ‘The free drug could be aggregated and formed pellet due to their hydrophobicity.’ and the reference are added in a revised manuscript.

Reviewer 2 Report

The authors have prepared an active substance-loaded, coated balloon (DEB) that may be suitable for percutaneous coronary interventions (PCI). In the case of a system containing the active ingredient, it was prepared by multi-layer coating. The active substance (Everolimus) was pre-encapsulated in a liposome. Liposomes were produced by the thin layer technique. The theme is actual, the manuscript contains several studies (determination of physicochemical properties; ex vivo measurements) in addition to production. A well-written manuscript, however, needs to be supplemented at some points: 
- A 2.2. in part, the authors describe the preparation of liposomes. How long did the drying described in line 111 take? At what setting parameters was lyophilization performed? 
- It is missing exactly how much Lipo + EVL was applied to the surface of the balloon (μg/mm2
- In Section 2.3, the amount of liposome + EVL was measured according to the text of the manuscript for the HPLC measurements. What was exactly determined was the drug only, or Lipo + EVL, since, in section 2.8, the drug was already measured by HPLC. 
-Is the same HPLC method used in sections 2.3 and 2.8? 
-Not indicated in 2.3. the column temperature. 
- The in vitro release study needs to be clarified. What equipment was used? What was the volume of the release medium? Has the volume taken for sampling been replaced with fresh buffer? 
-Figure 2 (E) lacks an indication of what a circle, square, and triangle mean.
- Fig. 2. (A): Indication of ratios is missing. 

Reviewer 3 Report

Dear authors, 

The article is truly interesting for the scientific community, with scientific soundness. The application is well demonstrated and the characterization is complete and well described. Results are well discussed and referenced. This paper has some problems with the English redaction, I recommend that you should ask a native English speaker to revise the manuscript. conclusions and references are adequate

Some minor comments are enlisted below

a) I found some typing errors (lines 23, 398), but revise carefully the whole manuscript again

b) Can you please, review this idea, and improve it "Especially, local drug delivery by way of DESs in which a drug such as proliferation inhibitor or immunosuppressant to reduce the rate of restenosis after PCI, one of the major drawbacks associated with bare metal stents, has shown to be a successful approach for CAD" I had to read it twice to understand it (lines 39-41)

c) Can you please separate in shorter paragraphs the introduction and discussion sections? it's difficult to read.

d) Please use the same format for "units". In some places I found "ml" and others "mL". Please revise the manuscript carefully and correct it

e) Please make sure that all SEM and fluorescence microscopy images have their magnitude size line (Figure 2A; Figure 3C; Figure 4A and B; Figure 5D and E; Figure 6B; Figure 7C)

f) Please eliminate one of the two repeated words: "In this study, the Nylon tube (same components as balloon catheter) was utilized for efficient experiments in this study." (lines 435, 436)

Round 2

Reviewer 2 Report

Dear Authors, 

Dear Editor,

Thank you for your answers. I have reviewed the corrections and accept them. I have no more questions and comments for the manuscript.